# AxiomVision: Accuracy-Guaranteed Adaptive Visual Model Selection for Perspective-Aware Video Analytics

Xiangxiang Dai
The Chinese University of Hong
Kong, Hong Kong, China
xxdai23@cse.cuhk.edu.hk

Zeyu Zhang*
Huazhong University of Science and
Technology, Wuhan, China
zeyuzhangzyz@gmail.com

Peng Yang
Huazhong University of Science and
Technology, Wuhan, China
yangpeng@hust.edu.cn

Yuedong Xu
Fudan University,
Shanghai, China
ydxu@fudan.edu.cn

Xutong Liu†
The Chinese University of Hong
Kong, Hong Kong, China
liuxt@cse.cuhk.edu.hk

John C.S. Lui
The Chinese University of Hong
Kong, Hong Kong, China
cslui@cse.cuhk.edu.hk

## Abstract

The rapid evolution of multimedia and computer vision technologies requires adaptive visual model deployment strategies to effectively handle diverse tasks and varying environments. This work introduces *AxiomVision*, a novel framework that can guarantee accuracy by leveraging edge computing to dynamically select the most efficient visual models for video analytics under diverse scenarios. Utilizing a tiered edge-cloud architecture, *AxiomVision* enables the deployment of a broad spectrum of visual models, from lightweight to complex DNNs, that can be tailored to specific scenarios while considering camera source impacts. In addition, *AxiomVision* provides three core innovations: (1) a dynamic visual model selection mechanism utilizing continual online learning, (2) an efficient online method that efficiently takes into account the influence of the camera's perspective, and (3) a topology-driven grouping approach that accelerates the model selection process. With rigorous theoretical guarantees, these advancements provide a scalable and effective solution for visual tasks inherent to multimedia systems, such as object detection, classification, and counting. Empirically, *AxiomVision* achieves a 25.7% improvement in accuracy.

## CCS Concepts

• **Information systems** → **Multimedia streaming**; • **Theory of computation** → *Online learning algorithms.*

## Keywords

Video analytics, model selection, online learning, camera perspective, neural networks

**ACM Reference Format:**
Xiangxiang Dai, Zeyu Zhang, Peng Yang, Yuedong Xu, Xutong Liu, and John C.S. Lui. 2024. AxiomVision: Accuracy-Guaranteed Adaptive Visual Model Selection for Perspective-Aware Video Analytics. In *Proceedings of the 32nd*

*Work conducted during Zeyu Zhang's visit to The Chinese University of Hong Kong.
†Xutong Liu is the corresponding author.

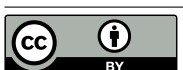

*MM '24, October 28-November 1, 2024, Melbourne, VIC, Australia*
© 2024 Copyright held by the owner/author(s).
ACM ISBN 979-8-4007-0686-8/24/10
https://doi.org/10.1145/3664647.3681269

*ACM International Conference on Multimedia (MM '24), October 28-November 1, 2024, Melbourne, VIC, Australia.* ACM, New York, NY, USA, 10 pages. https://doi.org/10.1145/3664647.3681269

## 1 Introduction

Video analytics plays a pivotal role in a multitude of tasks in a smart city, including vehicle license tracking, facial recognition, and traffic monitoring [25, 42]. This variety of applications highlights the necessity for customized visual models designed to cater to the unique requirements of different visual tasks. Yet, the application of such models, particularly those based on deep neural networks (DNNs), faces formidable challenges. These include the highly diverse requirements of video analytics tasks, fluctuating environmental conditions, and the imperative for real-time operation [7, 8]. The complexity and computational intensity of cutting-edge visual models in multimedia systems further complicate their application in resource-limited settings [41, 75]. Bandwidth constraints, for example, limit the feasibility of transmitting high-resolution video for analysis [77], highlighting a bottleneck in the practical utility of these complex technologies.

The **first challenge** in video analytics centers on *which visual model to apply that caters to application-specific requirements under a dynamic environment*, especially in light of phenomena like *data drift* [6, 10], where live video data's characteristics stray from the training dataset, as demonstrated in our Section 2. Despite a wide range of advancements targeting various specific requirements, efforts to modify existing models or customize visual models remain predominantly focused on the static and single scenario. For instance, works such as model pruning and compression [20, 30, 55], while effective at streamlining complex models for resource-limited environments, face significant performance degradation under adverse conditions like poor lighting or extreme weather [22, 31]. Given the above analysis, we advocate for *the strategic combination of existing models to navigate the complexities of real-world scenarios, instead of solely relying on a singular model or the pursuit of a one-size-fits-all visual model for universal video analytics.*

The complexity further escalates with the **second challenge**, as the video analytic system transitions from single-camera sources to multi-camera feeds. Independent decision-making for each camera regarding model selection would cause the computational load to increase linearly, which is unsustainable and retards the model selection process. Strategies such as the "*follow-the-leader*" [28],

*Spatula*'s camera correlation prioritization [27], and the *CrossRoI* system's re-identification algorithm [18] offer potential solutions for managing multi-camera setups. Nevertheless, these offline, periodically preset camera groupings, which depend exclusively on static search clustering and similar approaches, lack the essential flexibility needed to adapt to the dynamic nature of real-world environments and the process of continual learning for improvement.

Moreover, the significance of camera deployment in video analytics has been largely overlooked, with attention primarily focused on enhancing visual models on the server side, whether in the context of automated surveillance or user-controlled VR cameras. Nevertheless, for specialized tasks such as the creation of holographic stereogram portraits [14, 35], it is essential to investigate the additional effects arising from different camera perspectives, particularly when the perspective is altered. Additionally, due to declining costs, the popularity of cameras with adjustable viewpoints has surged (for instance, in 2020, the global market value of pan-tilt-zoom (PTZ) cameras reached $3 billion [26, 66]). Current methodologies, such as configuration adjustments in inference settings [28, 73], optimizing encoding [62, 77], and filtering out superfluous details [12, 40], presuppose an immutable scene captured by cameras. However, the variation of camera contents, application-specific requirements, and adjustable perspective necessitate moving beyond exclusively relying on pre-established offline learning methods, to understand the effects of camera perspectives for the accurate selection of visual models (the **third challenge**).

To address these challenges, this paper introduces *AxiomVision*, a novel framework designed to guarantee the accuracy of video analytics through the dynamic selection of visual models. Contrary to systems limited to data center or cloud environments [60, 71] that introduce issues like increased central load and the risk of congestion [68], *AxiomVision* leverages edge computing [68, 76] to decentralize processing, utilizing efficient lightweight DNN models which are deployed close to data sources. With its tiered edge-cloud architecture, *AxiomVision* strikes a balance between leveraging currently effective visual models and exploring promising, yet untapped models. This is achieved by continually analyzing observation feedback that includes camera perspective effects, even in the absence of prior knowledge about these perspectives. Furthermore, recognizing the potential for correlation among camera groups, *AxiomVision* incorporates a graph-based grouping method based on the natural camera network topology. This approach enables flexible and continual adjustment for camera groups for various visual tasks. To conclude, the manners in which this paper tackles the aforementioned third challenges can be summarized as follows:

**C1, Dynamic Visual Model Selection.** We enhance task-specific visual performance by employing an "*online learning strategy*" to select the optimal visual model dynamically. This method is different from previous work that relies solely on a single model or focuses on model enhancement. By incorporating continual feedback, our strategy employs a dynamic selection mechanism to identify the best-suited model adaptively. This mechanism is based on a tiered edge-cloud architecture, which is designed for deploying a diverse range of visual models, thus ensuring a wide selection availability.

**C2, Camera Network Topology Utilization.** Recognizing the common practice of deploying cameras in groups, we leverage the inherent network topology of these groups to develop a group-based mechanism to expedite the model selection process, especially in scenarios where there is no clear data to determine the optimal model or the impact of perspective. We demonstrate that this approach significantly alleviates the demands of continual learning, streamlining the operation in grouped camera environments.

**C3, Camera Perspective Consideration.** In response to the increasingly adjustable function of the modern camera [66] and our observation of measurement, we develop a "*perspective-aware learning method*" for cameras. This method goes beyond the conventional approaches which merely focus on improving visual models, but we uniquely account for the impact of the source-side model selection, namely, "*camera perspectives*", through online sensing estimation during the visual model selection process.

Our code is publicly accessible at: Code Link. Additionally, this work does not raise any ethical issues.

## 2 Background and Motivation

We start with a discussion on related works. Then we present our experimental results to illustrate how visual models' performance varies under external environmental conditions and across different visual tasks, underscoring the necessity of dynamic adaptation. Finally, we present the effects of the camera perspective.

### 2.1 Related Work

In video analytics, video frames are continuously streamed from one or more cameras to servers for processing. This often involves multiple visual models to support a multitude of video analytics applications, particularly those based on diverse architectures and weights of DNNs enabling them to accommodate an extensive array of scenes and vision tasks [18, 53]. However, the growing complexity of DNN architectures has resulted in increased prediction latency, presenting a considerable challenge for resource-limited end devices [39, 61]. To handle this, significant efforts have been devoted to leveraging lightweight models with streamlined architectures and fewer parameters [20, 74]. Yet, despite their efficiency under specific conditions or in tailored environments, these lightweight models often fall short in dynamic object distributions or challenging environmental conditions. Furthermore, the high costs of dynamically retraining model methods [6, 31] for specific scenarios make real-time maintenance challenging under changing conditions [21]. Faced with this, our method focuses on how to adapt to dynamic environments through the prioritization of continuous online visual model selection, moving away from the reliance on a few static models. More importantly, we demonstrate the importance of "*camera perspectives*" on the model selection.

Regarding model selection, we implement the strategy derived from multi-armed bandit (MAB), which performs online section of one or more options from a set of alternatives based on feedback from previous choices [34]. MAB has been widely applied in various domains, including recommendation systems [11, 23], content delivery [46, 69], and DNN design [70, 72]. Although some previous works on clustering bandits have explored grouping human users [17, 38, 44, 65], its application in the intricate domain of machine-centric video analytics remains under-explored, which uniquely focuses on maximizing inference accuracy and handling issues such as frame drops, provided the analytics' integrity is maintained [73].

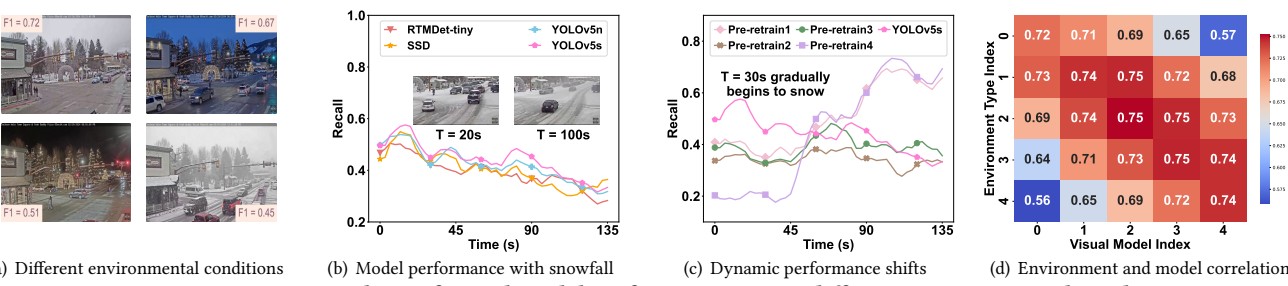

(a) Different environmental conditions    (b) Model performance with snowfall    (c) Dynamic performance shifts    (d) Environment and model correlation

**Figure 1: Comparative analysis of visual model performance across different environmental conditions.**

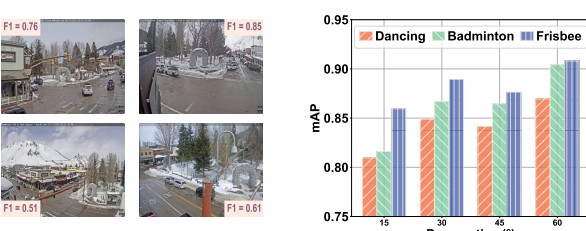

(a) Variability in perspective    (b) Varying segmentation accuracy

**Figure 2: Role of camera perspective in object detection and semantic segmentation visual tasks.**

## 2.2 Motivation Experiments

To investigate the variability of various models across different scenarios and the impact of camera perspectives, we conduct a thorough comparative analysis encompassing a range of models such as the YOLOv5 series [29], RTM detection [50], SSD [43], and Faster R-CNN [54]. In particular, object detection and semantic segmentation are selected as the visual tasks in our experiments.

**Dataset.** To assess the performance of visual models under different environmental conditions, we compile five representative video datasets, each covering a specific real-world scenario. These datasets are sourced from publicly available videos on YouTube, identified by searching for specific keywords (e.g., "live stream webcams") and selecting those pertinent to traffic conditions. To explore the effects of varying camera perspectives, four videos with distinct perspectives are selected at the same time from the above sources [56–59]. Furthermore, the free-viewpoint videos are also utilized for evaluating perspectives, where videos of various human movements on an indoor stage are recorded using 12 cameras positioned at equal perspective intervals [19].

**Evaluation Metrics.** For object detection, performance is assessed using the recall and F1 score metrics. We run the YOLOv5-x model as the ground truth, with the detection confidence score set to 0.25, and the intersection over the union threshold for calculating recall set at 0.5 aligning with [15, 71]. For semantic segmentation, pixel accuracy is applied as the metric, and the complex PP-HumanSegV1-Server [47] model is run as the ground truth [48].

**Performance Variability of Visual Models.** Fig. 1(a) reveals image examples across daytime, nighttime, snowy, and dusky scenarios. The F1 scores of YOLOv5-n demonstrate fluctuations across these four environments. In Fig. 1(b), a 135-second video depicting a sudden heavy snowfall is analyzed, revealing a decline in performance across all models as the intensity of the snowfall increases. The images on T = 20s and T = 100s show a clear difference before



(a) Perspective 1 (0°)    (b) Perspective 2 (30°)    (c) Perspective 3 (60°)

**Figure 3: Semantic segmentation across diverse camera perspectives for the same dancer.**

and after the snowfall. Figs. 1(a) and 1(b) effectively illustrate our argument with extensive examples across diverse environments and utilizing various visual models: *a single universal model faces significant challenges when attempting to perform consistently in dynamic environments.* The models show performance fluctuations of varying magnitudes depending on the environmental conditions.

Furthermore, we pre-retrain the YOLOv5-s model on a snowy-day traffic road dataset[52] using four different learning rates and training parameters under 100 epochs. we apply these models to a scenario where snowfall gradually begins at 30 seconds and intensifies by 45 seconds. As illustrated in Fig. 1(c), models pre-trained for snowy conditions show improved performance as the snowfall increases. However, their performance under normal weather conditions is inferior to that of the standard YOLOv5-s model, likely due to overtraining on snowy data. In Fig. 1(d), we further pre-train models on the COCO dataset for 100 epochs under various lighting conditions and evaluate them using the same YouTube dataset. Although models specifically pre-trained for certain environments show enhanced performance, performance fluctuations also exist (the second row). Moreover, the challenge of accurately quantifying light levels in dynamic real-world environments complicates the direct matching of these conditions with an appropriate visual model. This underscores *the complexity involved in dynamically adapting visual models to suit changing environmental conditions.*

**Impact of Camera Perspective.** We now turn our attention to the significant effects of camera perspective on model selection. Initially, we evaluate the F1 scores for object detection tasks using videos taken from different perspectives at the same traffic intersection at the same time. Fig. 2(a) reveals that the F1 scores *significantly vary with the camera's viewpoint.* For instance, a direct frontal view achieves an F1 score of 0.85, which drops to 0.61 when the camera is positioned laterally. Furthermore, for the semantic segmentation task, we explore the effects of camera perspective through multi-angle videos of activities such as dancing, playing badminton, and throwing a frisbee in an indoor environment, with a specific example of dancing showcased in Fig. 3. According to Fig. 2(b), a frontal capture of a dancer yields a segmentation accuracy

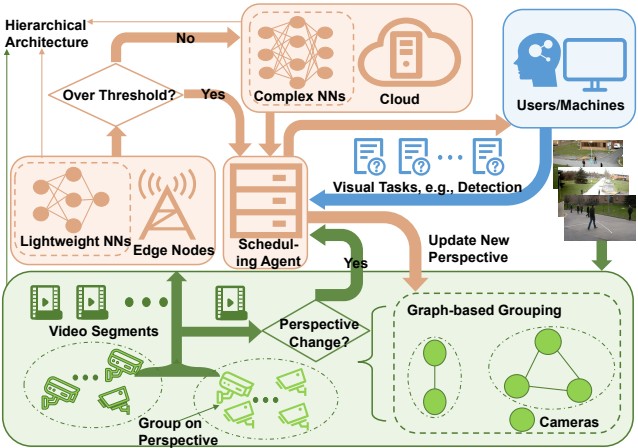

**Figure 4: Overview of *AxiomVision* framework.**

of 0.81, which surprisingly increases to 0.87 from a side angle. Additional analysis of two other movements shows similar patterns of accuracy variation. *The intrinsic relationship between perspective and model selection lies in the fact that certain perspectives may pose challenges for a task, e.g., distant blurred perspective in object recognition, requiring the use of more sophisticated models, whereas other perspectives can be addressed using simpler visual models* (Another example is in Appendix B). Therefore, the influence of perspective on model selection is a crucial factor that must be considered.

## 3 Model and Problem Formulation

In this section, we present the system model and the problem setting of our proposed *AxiomVision* framework. Our framework focuses on the adaptive selection of visual models under dynamic environmental changes and specific task demands, which utilizes a hierarchical architecture for camera groups, edge nodes, and cloud resources, and a continual video analytics pipeline to enhance visual task performance. Fig. 4 depicts our overall framework design.

### 3.1 System Model

**Tiered Edge-Cloud Architecture.** Traditional methods typically put all visual tasks, denoted as $Q$, to the cloud for centralized processing, which results in higher loads and longer response times on the cloud servers. To address these issues and also for the deployment of multiple visual models, we propose a hierarchical architecture, which consists of three levels: (a) the video server, (b) the edge nodes, and (c) the end camera groups. *AxiomVision* introduces a tailored combination set $\mathcal{M}_q$ of visual model models for each distinct visual task $q \in Q$ to better accommodate external variables such as lighting conditions and movement dynamics. Taking the object detection task as an example, lightweight models such as MobileNet [24] are implemented on the resource-limited edge nodes [78], while the deployment of more sophisticated models is designated for the cloud center. The total set of cameras in our system is represented by $\mathcal{N}$ with cardinality $|\mathcal{N}|$.

**Online Video Analytics Pipeline.** Our system is designed to adeptly handle the non-continuous and varied visual tasks through a sequential, discrete-time round approach. To accommodate the

varied nature and frequency of task demands, we define the sequence of rounds for each visual task $q \in Q$ as $\mathcal{T}_q$. For a specific visual task like facial temperature recognition, the duration between consecutive rounds may differ due to customer flow rates. Similar to [68], we assume that individual tasks do not interfere with one another, permitting each to operate in its designated round independently. Within this, the operational cycle of the online video analytics pipeline at each round involves the scheduling agent selecting visual models, obtaining feedback on their outcomes, and subsequently updating the evaluations for these models.

**Visual Payoff Feedback.** To dynamically adapt to evolving conditions and task-specific demands, we propose a visual payoff feedback mechanism, where *payoff* can be interpreted as the likelihood of meeting certain criteria, such as accuracy, recall, or F1 score. This process mainly focuses on the adjustment of perspective weights $\theta_n$ for each camera $n \in \mathcal{N}$ in response to task feedback of the selected visual model, thus enabling the online optimization of visual model efficacy.[1] For any given task $q \in Q$, the expected payoff $r_{m_t,t}$ of the selected visual model $m_t$ is expressed as:

$$\mathbb{E}[r_{m_t,t}|m_t] = \mu(\mathbf{x}_{m_t}^\top \boldsymbol{\theta}_n), \tag{1}$$

where $\mu$ establishes a nonlinear connection between the payoff $r_{m_t,t}$ and the feature vector $\mathbf{x}_{m_t}$ of the visual model $m_t$ at round $t$, incorporating the influence of the camera perspective through the weight vector $\theta_n$ [16, 32]. An example of such a link function is a neural network, wherein a final layer equipped with either a sigmoid or ReLU activation function transforms the intricate features derived into meaningful results [33, 67].

**Combinatorial Model Selection.** Unlike conventional works that may only offer a single model choice per task, we develop a "*combinatorial model selection strategy*" from a wide range of model candidates, which increases the probability of meeting the task's requirements under diverse environmental conditions. Specifically, a set of visual model options $\mathcal{M}_t = \{m_1, \ldots, m_{|\mathcal{K}_t|}\} \subseteq \mathcal{M}_q$ is presented for each task $q$ at every round $t$. $\mathcal{K}_t$ with the size of $|\mathcal{K}_t|$ represents the selected visual model index at round $t$, determined by the first selected model with $r_t = 1$. Here, if the accuracy of the selected visual model surpasses the predefined threshold, the resulting payoff value $r_t$ will be assigned a value of 1. Initially, to conserve resources and ensure rapid response, priority is given to models deployed at the edge. If these models fall short of the task's required accuracy threshold, the scheduling agent will choose more complex models in the cloud. More importantly, the selection of visual models is continually refined based on payoff feedback. Thus, at each round $t$, the aggregate payoff from these combinatorial usages of visual models $\mathcal{M}_t$ for camera $n$ is calculated as:

$$R(\mathcal{M}_{n,t}) = 1 - \prod_{k=1}^{|\mathcal{K}_t|} (1 - r_{m_{k,t},t}(m_{k,t})), \text{ where } k \in \mathcal{K}_t, m_{k,t} \in \mathcal{M}_t.$$

Note that we provide a combination of visual models based on the tiered architecture to ensure accuracy. Moreover, we also strive to ensure that the initially chosen visual model meets the requirements as much as possible (see Section 4 for details).

---

[1]For notation clarity, we initially focus on the impact characteristics related to model selection from a singular camera perspective. Nonetheless, our forthcoming strategy for evaluating and categorizing perspective impacts is readily scalable to accommodate the diverse impact characteristics of changeable camera perspectives (see Section 5).

**Perspective-Based Grouping.** For visual tasks that necessitate inputs from multiple cameras, such as person tracking, the process entails more than just evaluating how camera perspectives influence the choice of visual models. It also presents a potential opportunity to group cameras based on their overlapping perspective weights, where the increasing deployment of camera network topologies inherently leads to a common occurrence of overlapping perspectives [9]. Recognizing this, we propose to group cameras with similar influence of perspective $\theta_n$ to adopt a similar visual model selection. To distinguish cameras that cannot be grouped together, we evaluate the dissimilarity between cameras using the distance between the feature vectors representing camera perspective effects. This evaluation involves establishing a "dispersion" criterion: cameras $j, k$ are considered for separate groups if:

$$\|\theta_j - \theta_k\|_2 \geq \gamma_q, \forall j, k \in \mathcal{N}, \tag{2}$$

where $\gamma_q$ signifies a predetermined positive dispersion constant specific to task $q$. Based on the above criterion, the collective set of cameras $\mathcal{N}$ can be divisible into smaller subsets, and we label them as $G_1, G_2, \ldots, G_g$, wherein cameras within the same subset adhere to a unified visual model selection strategy. Note that neither which camera belongs to which group, nor the precise number of grouped cameras $g$ can be not known beforehand in our model.

## 3.2 Problem Formulation

The selection of an optimal visual model is influenced by the diverse visual tasks, external environment, and internal deployment factors, namely camera perspectives. As a result, the scheduling agent must continually adapt the choice of visual model $m_t \in \mathcal{M}$ for each processed camera $n_t \in \mathcal{N}$ at the round $t \in \mathcal{T}_q$ for visual task $q$, in alignment with the video analytics payoff. In this work, we propose to dynamically adapt visual models in an online manner to maximize the overall payoff across all rounds for any given visual task $q \in Q$. This objective is mathematically expressed as:

$$\max \mathbb{E}\left[\sum_{t \in \mathcal{T}} \sum_{n \in \mathcal{N}} \mathbf{1}\{n_t = n\} R(\mathcal{M}_{n_t, t})\right], \tag{3}$$

where $\mathbf{1}\{\cdot\}$ denotes the indicator function. Given the limited bandwidth and computing resources, it is not possible to select the most resource-intensive visual model option [28]. Furthermore, with the growing deployment of extensive camera networks by various entities, an exploration into the impact of perspective similarity among cameras is needed. By analyzing these perspective weight similarities, we aim to identify and group cameras with similar perspective influences, thereby reducing the number of subgroups needed for visual model sharing. Nevertheless, this endeavor introduces several challenges, including complex search spaces, varying effects of camera perspectives, zero prior knowledge of optimal visual models, and the strategic deployment of camera groups. Addressing these issues requires an adaptive algorithm capable of learning and adjusting visual models online for diverse visual tasks, transcending the limitations of static, offline model selection strategies.

## 4 Continual Learning of *AxiomVision*

In this section, we first present the algorithm design of *AxiomVision*, followed by a performance analysis. Specifically, a flexible graph-based structure is utilized to mirror the natural undirected connectivity found in camera cluster networks.[2]

## 4.1 Algorithm Design

---
**Algorithm 1** Continual Online Learning of *AxiomVision*

---
**Require:** Set of cameras $\mathcal{N}$; Parameter $\alpha, \beta$; Random $p_0 \in (0, 1)$.
**Ensure:** Visual model selection for all visual tasks.
1: **Initialization**: A complete graph $U_0 = (\mathcal{N}, E_0)$; $g_1 = 1$; $T_{n,0} = 0, \forall n \in \mathcal{N}$.
2: **for** each $q \in Q, t \in \mathcal{T}_q$, **independently do**
3:     Receive processed camera index $n_t$;
4:     Identify group $G_{i_t}$ that contains $n_t$;
5:     Estimate perspective weight $\hat{\theta}_{i_t, t-1}$ based on Eq. (4);
6:     Select the combinotorial model set $\mathcal{M}_t \in \mathcal{M}_q$ according to Eq. (5) until the predetermined threshold is satisfied;
7:     Record payoff $r_{m_k, t}$ of the selected visual model $m_k, k \in \mathcal{K}_t$;
8:     Increment count of processed camera $n_t$: $T_{n_t, t+1} = T_{n_t, t} + 1$;
9:     Delete from $E_t$ all $(n_t, \ell)$ if Eq. (6) holds and get the resulting graph $U_{t+1} = (\mathcal{N}, E_{t+1})$;
10:    Update graph parameter: $p_t = p_0/t^2$;
11:    Reconnect all edges in $E_{t+1}$ with probability $p_t$;
12: **end for**

---

**Assigning Inferred Groups for Processed Cameras.** Initially, we employ some common clustering methods, e.g., $K$-means, to group cameras with similar perspective impact weights. Within each cluster, a fully connected graph is first established, reflecting camera connectivity symmetry and dynamic adaptability for network topology, facilitating continual online updates. In the absence of initial perspective impact weight information, the entire camera cluster can be initialized as an undirected fully connected graph $U_0 = (\mathcal{N}, E_0)$, maintained for visual task $q \in Q$, where each camera $m \in \mathcal{N}$ represents a node in the graph. Cameras sharing similar learned perspective impact weights are interconnected via edges in $E_0$. At each round $t$, the connected components within graph $U_t$ signify the inferred groups $G_1, G_2, \ldots, G_{g_t}$, with $g_t$ denoting the the number of camera groups at $t$. Initially, graph $U_1$ is a complete graph, with $g_1 = 1$. For the processed camera $n_t$, *AxiomVision* determines camera $n_t$'s group index $i_t$ by identifying $U_t = (\mathcal{N}, E_t)$ to find the group that camera $n_t$ belongs to, i.e., $n_t \in G_{i_t}$.

**Perspective-aware Weight Estimation.** To maximize the payoff as defined in Eq. (1), we propose to use the maximum likelihood estimator $\hat{\theta}_{i_t, t}$ to test whether cameras belonging to group $i_t$. This estimator is designed to yield a unique solution, expressed as:

$$\sum_{j=1}^{t-1} \mathbf{1}\{n_j \in G_{i_t}\} \sum_{k=1}^{|\mathcal{K}_j|} \left(r_{m_k, j} - \mu(x_{m_k, j}^\top \hat{\theta}_{i_t, t})\right) x_{m_k, j} = 0. \tag{4}$$

Eq. (4) represents the condition for optimality, where the cumulative discrepancy between the predicted payoff and actual payoff sums to zero, as estimated by the membership indicator of cameras within the same group $i_t$. We use Newton's method which allows for efficient computation of the solution [16, 38]. The historical

---
[2]For brevity, we focus on the algorithmic procedures for a singular visual task $q \in Q$, noting that the procedures can be executed in parallel for multiple visual tasks $Q$.

feedback data from all cameras in the same group, not just camera $n_t$, are used to update the estimation, emphasizing the value of grouping for process acceleration. For the cases where the exact value of $\mu$ is unknown, the process can be reduced to an estimation process using $x_{m_k,j}^\top \hat{\theta}_{i_t,t}$ (The subsequent visual model selection component following the same method).

**Selecting Visual Model with Optimistic Approach.** Following group assigning and estimation, the currently processed camera undergoes optimization and reward feedback observations on the selected visual model. Note that the greedy visual model selection strategy, i.e., $\text{argmax}_{m \in \mathcal{M}_q} \mu(x_m^\top \theta_{n_t})$, might result in insufficient exploration of undiscovered visual models, thereby failing to ensure optimal model selection. We propose to address this challenge by adopting an "*optimistic approach to encourage exploration*" among different visual models [34, 63]. Specifically, for any processed camera $n \in \mathcal{N}$, we define the Gramian matrix as $M_{n,t} = \sum_{\substack{j \leq t \\ n_j=n}} \sum_{k=1}^{|\mathcal{K}_j|} x_{m_k,j} x_{m_k,j}^\top$, and for the belonging group index $i$ of camera $n$, denote $M_{i,t} = \zeta I_d + \sum_{n \in G_i} M_{n,t}$, where $\zeta I_d$ is a regularization term added to improve stability. Based on the estimated $\hat{\theta}_{i_t,t-1}$ for group $G_{i_t}$, the visual model for round $t$ is selected via the upper confidence bound strategy:

$$m_t = \underset{m \in \mathcal{M}_q}{\text{argmax}} \left( \mu(x_m^\top \hat{\theta}_{i_t,t-1}) + \alpha \|x_m\|_{M_{i_t,t-1}^{-1}} \right), \quad (5)$$

where $\|x\|_M := \sqrt{x^\top M x}$ and $\alpha$ is a positive parameter. Note that Eq. (5) incorporates *both* empirical payoff exploitation from the first term, as well as exploration of different visual models through the upper confidence bound. The observed payoff $r_{m_t,t}$ is then recorded so to update the evaluated performance of visual model $m_t$.

**Optimizing Selection for Adaptive Accuracy.** Subsequently, the agent selects a set of visual models $\mathcal{M}_t = (m_1, \ldots, m_{|\mathcal{K}_t|})$ for the processed cameras using the above optimistic selection strategy. It initially prioritizes visual models based on Eq. (5), by consideration of their deployment on edge nodes and in the cloud. If the accuracy does not meet the predetermined threshold, the process continues with the next model, until a stopping criterion is met. This strategy enables us to prioritize potentially high-performing and lightweight models while ensuring accuracy even under unforeseen circumstances, such as sudden snowy weather. Additionally, the scheduling agent updates its estimation of visual models based on the received payoff feedback of each selected visual model, to ensure that subsequent selections meet the accuracy requirements on the first attempt as much as possible. Through this approach, our proposed method can achieve rapid visual model adaptation while ensuring adaptive accuracy across different conditions.

**Updating Dynamic Graph for Grouping.** For any processed camera $n \in \mathcal{N}$, we define $T_{n,t} = \sum_{\substack{j \leq t \\ n_j=n}} |\mathcal{K}_j|$ as the number of effective feedbacks up to round $t$. The dynamic graph structure is then updated to reflect changes in camera grouping, particularly adjusting based on the current inferred similarity in perspective weights. An edge $(n_t, \ell)$ is removed if:

$$\left\| \theta_{n_t,t-1} - \theta_{\ell,t-1} \right\|_2 > \beta \left( f(T_{n_t,t-1}) + f(T_{\ell,t-1}) \right), \quad (6)$$

with $f(x) = \sqrt{\frac{1+\log(1+x)}{1+x}}, x \geq 0$. This deleting function stems from a theoretical optimal graph structure threshold, modified here

for computational feasibility while still maintaining theoretical validity [17]. Further comparisons will be illustrated in Section 5. The updated graph $U_t$ is utilized in the subsequent round.

**Adaptive Graph Reconstruction Strategy.** To avoid mistakenly removing edges that might be correct, the scheduling agent reinstates the undirected complete graph at a certain probability. As more payoff feedback is gathered, the accuracy of the estimated groupings improves the perspective influence on visual models. Consequently, there is a diminished need for frequent graph reconstructions, and this motivates us to have a design where the probability $p_t$ decreases over time. Initially, $p_0$, set within the range (0,1), is determined randomly (Line 10-11).

Note that our continual online design *complements*, rather than competes with traditional offline methods. For example, if prior knowledge exists about the effects of camera perspectives or the choice of models via the offline methods, it can be easily assimilated into our strategy and progressively refined based on *AxiomVision*.

## 4.2 Performance Analysis

For the ease of presenting our theoretical analysis, let $\|x_{m_t}\|_2 \leq 1$ and $\|\theta_{n_t}\|_2 \leq 1, m_t \in \mathcal{M}_q$ for all rounds. At each round $t$, a camera is randomly processed from $\mathcal{N}$ with uniform probability for fairness, independently of selections in previous rounds. For every visual task $q \in Q$, we evaluate our algorithms by measuring the greatest difference in payoff between the theoretically ideal visual model (not known beforehand) and the visual model actually chosen. This difference is defined as "*regret*" [5, 34, 45], expressed as:

$$Reg(T_q) = \mathbb{E}\left[ \sum_{t \in \mathcal{T}_q} \sum_{n \in \mathcal{N}, n_t=n} R(\mathcal{M}_{n_t}^*) - R(\mathcal{M}_{n,t}) \right], \quad (7)$$

where $T_q$ denotes the cardinality of $\mathcal{T}_q$ and $\mathcal{M}_{n_t}^*$ denotes the "*unknown*" optimal combinatorial set of visual models for task $q$.

In line with [17, 38, 44], we posit that $\mathbb{E}\left[x_{m_t} x_{m_t}^\top\right]$ is full rank, with a minimum eigenvalue $\lambda > 0$, and that $x_{m_t}^\top \theta_{n_t}$ exhibits a sub-Gaussian tail with a variance not exceeding $\sigma^2$. Furthermore, following [16, 38], we consider $\mu$ to be a strictly increasing, continuously differentiable link function that is Lipschitz continuous with constant $L$. We denote $m_\mu = \inf_{a \in [-2,2]} \mu'(a)$ and assume $m_\mu > 0$. Defining $\tilde{\lambda}$ as the integral $\int_0^\lambda (1 - e^{-\frac{(\lambda-x)^2}{2\sigma^2}})^K dx$ with $K$ indicating the maximum number of selected combinatorial visual models across all rounds [64], we set the tuning parameters $\alpha$ and $\beta$ as follows: $\alpha = \frac{1}{m_\mu}\sqrt{\frac{8}{\tilde{\lambda}} + d\ln(T/d) + 2\ln(4gT)}$ and $\beta = \sqrt{32d/(\tilde{\lambda}m_\mu^2)}$, where $d$ and $g$ represent the dimension of the vector and the maximum number of camera groups under all adjustable perspectives, respectively. Then, we give the following performance guarantee.

**THEOREM 1** (REGRET UPPER BOUND). *The regret of AxiomVision throughout $\mathcal{T}_q$ is bounded by $Reg(T_q) \leq O\left(\frac{Ld}{m_\mu}\sqrt{gKT_q}\ln(T_q)\right)$.*

**Remark:** Theorem 1 suggests that the payoff from video analytics can approach near-optimal performance asymptotically over rounds, signifying that $\lim_{T_q \to \infty} \frac{Reg(T_q)}{T_q} = 0$. The expected regret of video analytics payoff, defined in Eq. (7), arises from two primary factors for a given visual task $q$: the rounds needed to gather sufficient information for accurate camera attribute estimation and grouping, and the practice of sharing the visual model within the

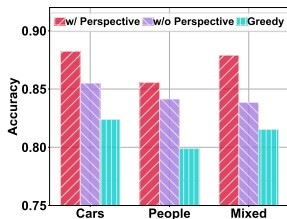
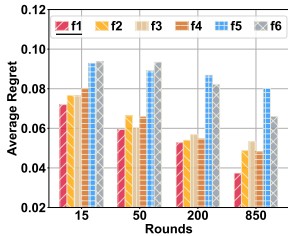
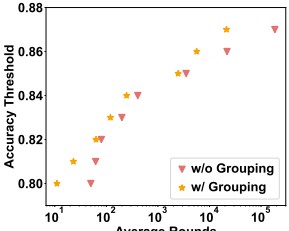
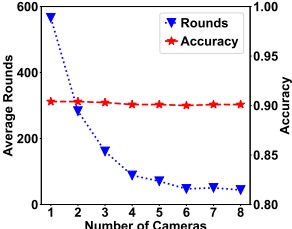

**Figure 5: Comparison when perspective is not considered.**

**Figure 6: Different deleting function $f(x)$ on regret.**

**Figure 7: Influence of grouping cameras on acceleration.**

**Figure 8: The benefit of increased camera count.**

same group instead of making independent selections. Compared to the ideal scenario where camera grouping is known and cameras have equal adjustable perspectives, the theoretical convergence of *AxiomVision* is nearly optimal. This scenario is analogous to managing $g$ independent groups, each undergoing $T_q/g$ learning rounds, resulting in a regret lower bound of $\Omega(\frac{L}{m_\mu}\sqrt{dgT_q})$ [13].

**Proof Sketch:** For any camera $n$ with group index $i$, denote the frequency associated with group $G_i$ up to round $t$ as $T_{i,t} = \sum_{n \in G_i} T_{n,t}$, and $g_{i_t,t}(\theta) = \sum_{j=1}^{t-1} \mathbf{1}\{n_j \in G_{i_t}\} \sum_{k=1}^{|\mathcal{K}_j|} \mu(x_{m_k,j}^\top \theta) x_{m_k,j}$. With Eq. (4), then $g_{i_t,t}(\hat{\theta}_{i_t,t-1}) = \sum_{s=1}^{t} \mathbf{1}\{i_s \in I\} \sum_{k=1}^{|\mathcal{K}_j|} r_{m_k,j} x_{m_k,j}$. With probability at least $1 - \delta$, for some $j \le t$ with $M_{i_t,j}$ invertible:

$$\left\| g_{i_t,t}(\hat{\theta}_{i_t,t}) - g_{i_t,t}(\theta_{i_t}) \right\|_{M_{i_t,t}^{-1}}^2 \le T_{i_t,j}\lambda_{\min}(M_{i_t,t})^{-1} + d \ln \frac{T_{i_t,t}}{d} + 2\ln\frac{1}{\delta}$$

Here, $\lambda_{\min}(M)$ denotes the minimum eigenvalue of matrix $M$. Then, $\boldsymbol{\alpha}(t,\delta) = \frac{1}{m_\mu}\sqrt{\frac{8}{\lambda} + d\ln\frac{t}{\lambda} + 2\ln\frac{1}{\delta}}$, by the property of Lipschitz, we can assert that the currently estimated group index $i_t$ for camera $n_t$ is correct. Consequently, the correct grouping can be formed based on the deleting rule of Eq. (6). Consider the instantaneous regret $Reg_t$ at round $t$ for the selected visual model $m_t$ under task $q$. Given the correct grouping, we obtain: $Reg_t = \mu(x_{m^\star}^\top \theta_{n_t}) - \mu(x_{m_t}^\top \theta_{n_t}) \le 2\alpha L \left\| x_{m_t} \right\|_{M_{i_t,t-1}^{-1}}$. Finally, we derive: $Reg(T_q) = \mathbb{E}\left[\sum_{t=1}^{T_{q,0}} \mathbb{E}_t(Reg_t)\right] + \mathbb{E}\left[\sum_{t=T_{q,0}+1}^{T_q} \mathbb{E}_t(Reg_t)\right] \le O\left(\frac{Ld}{m_\mu}\sqrt{gKT_q}\ln(T_q)\right)$. ∎

# 5 Performance Evaluation

## 5.1 Implementation and Setup

**Testbed.** Leveraging public 360° VR camera feeds from [1–4], our setup involves NVIDIA Jetson TX2, Nano, and TX2 NX end devices handling $|\mathcal{N}| = 308$ video segments from different perspectives, with ENs powered by NVIDIA GeForce RTX 4060 and HPC Dell PowerEdge R930 servers as the cloud center. Rectilinear images are extracted from panorama to function as adjustable perspectives [36]. Beyond the DNN model used in Section 2, we employ the lightweight YOLOv5-s, trained on the COCO dataset under diverse lighting, finally containing a total of 17 optional visual models. Visual tasks, in line with [40, 66], include *Classification*, *Counting*, *Detection*, and *Aggregation*. Utilizing approaches from [37, 44, 67], we construct and decompose a performance payoff matrix for these tasks across all video segments, extracting feature vectors for visual model index representation. Camera bandwidth varies between 1 and 2 Mb/s, with EN to server uplink around 10 Mb/s [25, 41].

**Metrics.** We evaluate the following performance metrics: (a) Accuracy: Assessed for the four visual tasks outlined in [40, 66]. (b)

Round: As described in Section 3. (c) Regret: Detailed in Eq. (7). (d) Time: Encompasses the algorithm's **execution time**, visual model **inference time** (e.g., YOLOv5-s), and initial **transmission time**. Notably, transmission and analysis are not sequential; initial data upload incurs a startup latency, followed by continuous and parallel transmission and analysis. (e) Bandwidth: Normalized bandwidth usage for transmitting encoded video segments. Through both theoretical and empirical analysis, *AxiomVision* parameters are set to $(\alpha, \beta) = (0.25, 0.1)$. Note that we periodically run YOLOV5-x to acquire true bounding boxes for accuracy assessment (included in the total consumption measurements), and additional extended experiments can be found in Appendix B.

## 5.2 In-depth Analysis of Exploring Results

In pursuit of evaluating the effectiveness and rationale of certain components within our *AxiomVision* design, we conduct a comprehensive series of experiments under the object detection task.

**Perspective Effects.** To underscore the importance of camera perspective alongside server-side models, we design an *AxiomVision* variant without perspective consideration, *w/o Perspective*, and compared it with a version integrating camera perspective, *w/ Perspective*, plus a greedy method providing fixed-perspective optimal model across all feeds. As depicted in Fig. 5, results show *w/ Perspective* improves mean accuracy by 2.7% over *w/o Perspective*. Moreover, both *w/o Perspective* and *w/ Perspective* by facilitating online model selection, surpass the greedy strategy by 2.3% and 5.6% in accuracy.

**Deleting Function Evaluation.** Based on [49, 51], we assess various deleting functions: $f_1(x) = \sqrt{\frac{1+\ln(1+x)}{1+x}}$ (ours), $f_2(x) = \frac{1}{(1+x)^2}$, $f_3(x) = \frac{1}{\sqrt{1+x}}$, $f_4(x) = \frac{1}{\sqrt[4]{1+x}}$, $f_5(x) = 1 + \ln(1+x)$, and $f_6(x) = \sqrt{1 + \ln(1+x)}$. We evaluate the regret incurred by different functions at 15, 50, 200, and 850 rounds for each camera, with the optimal strategy determined through the YOLOv5-x model. Fig. 6 shows that the function $\sqrt{\frac{1+\log(1+x)}{1+x}}$ consistently delivers optimal performance across various rounds with minimal regret.

**Grouping Impact on Acceleration.** Exploring the effect of camera grouping on acceleration within the *AxiomVision* framework, we compare performances between ungrouped (*w/o Grouping*) and grouped (*w/ Grouping*) setups, as illustrated in Fig. 7. By setting accuracy thresholds from 0.8 to 0.87, *w/ Grouping* significantly reduces the total number of rounds across all cameras by at least 1.27×, achieving an average acceleration of 3.23× and a median of 2.18×. Additionally, we observe that increasing the number of cameras leads to a reduction in the required rounds while achieving a similar level of accuracy, as shown in Fig. 8.

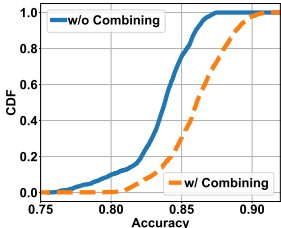

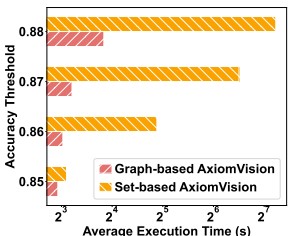

**Figure 9: Evaluating usage of combinatorial set of models.**

**Figure 10: Set-based vs. graph-based on executing time.**

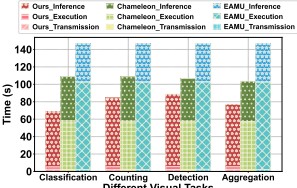

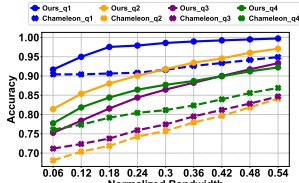

**Figure 12: Decomposition of total time overhead.**

**Figure 13: Accuracy with varying bandwidth.**

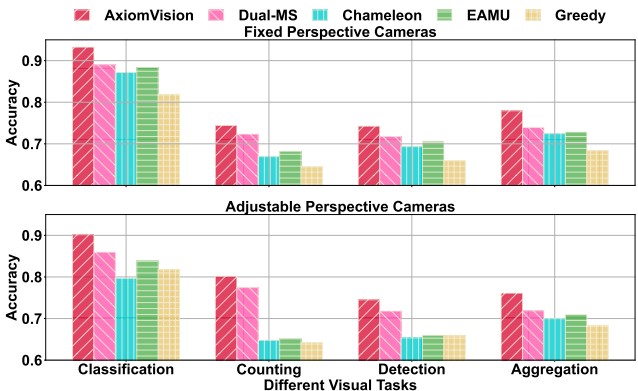

**Figure 11: Accuracy on fixed and adjustable perspectives.**

**Combinatorial Set Benefits.** Referring to Section 3, we address dynamic accuracy needs by assembling a combinatorial set of visual models for selection, as illustrated in Fig. 9. Highlighting this design's benefits, we compare it with a non-combinatorial system (*w/o Combing*), which, after applying *w/ Combing*, would run subsequent high-load models at equal probability. The cumulative distribution function (CDF) for accuracy of object detection in Fig. 9 shows that using the combinatorial set (*w/ Combing*) achieves a median precision improvement over the non-combinatorial method (*w/o Combing*) by 2.3%, with an average increase of 2.6%.

**Graph Grouping Efficiency.** In Algorithm 1, we implement a graph-based camera grouping strategy, and a set-based *AxiomVision* algorithm is designed here. Through testing execution time across various accuracy thresholds of 308 camera feeds, as depicted in Fig. 10, the graph-based approach significantly reduces execution time by factors of 1.13×, 3.62×, 9.82×, and 10.50×. Particularly in high-round scenarios, this graph-based grouping method notably surpasses the set-based grouping in time efficiency.

### 5.3 Benchmarking against State-of-the-Art

**Benchmarks.** Our comparison includes the following schemes. (1) *Chameleon*, capable of dynamically selecting the visual model based on temporal and spatial correlations [28]. (2) *Dual-MS*, inspired by [15], categorizes visual models into two layers: a simpler model and a more complex model, for effective model selection. (3) *EAMU*, standing for edge-assisted on model update in adverse environments [31]. (4) *Greedy*, which, during the initial short segment of analysis for all video sources offline, runs all models to identify the one offering the highest average accuracy.

**Accuracy on Fixed & Adjustable Perspectives.** The average accuracy of 2000 rounds is illustrated in Fig. 11. Across different visual tasks for fixed perspectives, *AxiomVision*, which employs a continuous online model selection, consistently outperforms *EAMU*, *Chameleon*, *Dual-MS* and *Greedy*. Furthermore, under adjustable perspectives, *EAMU* and *Chameleon* experience a deterioration in accuracy due to their lack of consideration about the impact of source-side camera perspectives.

**Decomposition of Total Time.** We compare our approach under $T_q = 2000, \forall q \in Q$ with *Chameleon* (setting its parameter interval = 5 and top-$k$ = 5 for re-profiling video pipelines, referred to as execution time), and *EAMU* (calculating its average training cost for retraining, also denoted as execution time). Fig. 12 indicate that although our method leads to an increase in inference time due to the adoption of a combinatorial design, it efficiently reduces execution time by eliminating the need for the re-profiling in *Chameleon* and the retraining process in *EAMU*. In comparison, *Chameleon* allocates nearly identical time for spatial-temporal profiling; *EAMU* incurs a significant additional time cost due to its retraining process. Moreover, the initiation time for transmission is markedly the smallest in scale under 200 kbps bandwidth constraint.

**Impact of Bandwidth Condition.** In Fig. 13, we benchmark our methodology against *Chameleon* across the above tasks (abbreviated as $q1, q2, q3, q4$). *EAMU* is omitted owing to its retraining architecture, which diverges from the context of bandwidth. The outcomes demonstrate that the strategic approach of *AxiomVision*, which involves selectively deploying complex models for tasks where accuracy is compromised, significantly boosts performance across all visual tasks. This advantage becomes particularly prominent in scenarios of limited bandwidth, underscoring our method's efficiency in bandwidth-restricted video analytics.

## 6 Conclusion

We propose *AxiomVision*, an innovative framework guaranteeing performance for a wide range of environments and visual tasks. *AxiomVision* leverages dynamic model selection and a tiered edge-cloud architecture. With experiments based on extensive real-world camera videos, *AxiomVision* introduces a novel approach to consider camera perspective and unveils a group-based acceleration strategy that capitalizes on camera cluster topology. Furthermore, *AxiomVision* is designed with a theoretical performance guarantee even under the worst-case scenarios, that is, *AxiomVision* can asymptotically converge to the optimal model section policy. Tested on a built platform, *AxiomVision* demonstrates superior performance over existing works, and greatly improves adaptability and efficiency across various video analytics applications.

## Acknowledgement

The work of Xiangxiang Dai and John C.S. Lui was supported in part by the RGC GRF 14202923. The work of Peng Yang was supported in part by the Natural Science Foundation of China under Grant 62001180, and in part by the Young Elite Scientists Sponsorship Program by CAST under Grant 2022QNRC001. The work of Yuedong Xu was supported in part by the Natural Science Foundation of China under Grant 62072117, and in part by the Shanghai Natural Science Foundation under Grant 22ZR1407000.

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
