# OpenReview forum: "AxiomVision: Accuracy-Guaranteed Adaptive Visual Model Selection for Perspective-Aware Video Analytics"
_acmmm.org/ACMMM/2024/Conference — MM2024 Poster_

### Official Review · Reviewer_Y6RN · 2024-04-29

**Rating:** 4
**Confidence:** 2

**Summary:**

The paper introduces a novel framework that dynamically selects the most efficient visual models for video analytics by leveraging edge computing. AxiomVision uses a tiered edge-cloud architecture to deploy a broad spectrum of visual models, from lightweight to complex deep neural networks, tailored to specific scenarios and considering camera source impacts. This approach is designed to ensure accuracy in visual tasks across varying environments and camera perspectives.

**Strengths:**

Clarity and Presentation: The paper is well-structured and clearly written, making complex concepts accessible. Diagrams and systematic descriptions enhance the clarity, facilitating understanding of the operational framework and its components.

**Limitations:**

1.Complexity and Scalability: The complexity of the AxiomVision framework, particularly in managing dynamic model selection across potentially numerous edge devices, could pose scalability challenges. The computational overhead and the coordination required for real-time updates might limit its deployment in extremely large-scale or resource-constrained environments.
2.Reference 39 lacks page numbers

**Suitability:**

2

---

### Official Review · Reviewer_pS5i · 2024-05-09

**Rating:** 3
**Confidence:** 3

**Summary:**

This paper presents AxiomVision, a novel framework that dynamically select the best models for video analytics under diverse scenarios. AxiomVision is designed to address three challenges: 1) model selection; 2) adopting to diverse application scenarios; 3) adopting to various camera perspectives.

**Strengths:**

1. Motivation is clear and easy to follow.
2. A mathematical problem is formulated and proved with performance analysis.

**Limitations:**

1. It is suggested that the author use either "video analysis" or "video analytics" instead of both throughout the paper, and make it consistent.
2. What does the value mean in Figure 1(d) mean? F1-score?
3. It is suggested that the author give more introduction about the graph network, and the rationale about why it is chosen as the solution here.
4. It seems the formulation and solving of the proposed problem is apart from the actual problem under the hood, i.e., addressing the challenges in scenario change, model selection and perspective change in edge-cloud video analytics setting. Section 5 is purely about the mathematical prove and it should be kept in the appendix without interfering the main system design.

A few typos:

1. Line 35, left column, "online learning" instead of "online larning"
2. Figure 1(d) caption, what is "Environment an model correlation"?

**Suitability:**

2

---

### Official Review · Reviewer_881f · 2024-05-24

**Rating:** 4
**Confidence:** 2

**Summary:**

This paper discuss the system design for perspective-aware video analytics. The authors propose dynamic visual model selection, camera network topology utilization and camera perspective consideration to tackle challenges. Some theoretical analyses are made, and experimental results show their effectiveness.

**Strengths:**

1. Strong motivation. There are clear analysis in the Introduction and Background part. Also some experiments are provided to address their concerns.
2. A reasonable method design. Using a graph-based approach sounds natural.

**Limitations:**

1. The writing is not good enough. For a reader not particularly familiar with the field, the relationships in chapters three, four and five are hard to sort out quickly.
2. The SOTA methods to be compared are not described in detail in the related work. Therefore, the differences between the methods cannot be directly compared.
3. Some ablation experiments or visualizations of the model selection method could be added to help readers understand the intuitive effect of this approach. For example, what will happen if 17 models trained under different lighting conditions are replaced by a model trained under 17 lighting conditions?

**Suitability:**

3

---

### Meta-Review · Area_Chair_sDDj · 2024-07-01

**Recommendation:** Accept (Poster)
**Confidence:** 4

**Metareview:**

The paper discusses the system design for perspective-aware video analytics, introducing AxiomVision, a novel framework designed to dynamically select the best models for video analytics in diverse scenarios. The framework addresses challenges related to model selection, adapting to various application scenarios, and considering different camera perspectives. This paper's strengths and limitations are listed as follows:
Strengths:
1. The paper clearly articulates its motivation with thorough analyses in the introduction and background sections.
2. It is well-structured and clearly written, making complex concepts accessible and using diagrams and systematic descriptions to enhance clarity.
3. The graph-based approach for model selection and camera network topology utilization is logical and well-justified, leveraging a tiered edge-cloud architecture to deploy a range of visual models tailored to specific scenarios and camera sources.
4. The paper provides a solid mathematical formulation of the problem, with performance analysis and experimental results demonstrating the effectiveness of the proposed method.

Limitations:

1. The relationships between chapters three, four, and five are hard to follow for readers not familiar with the field; the writing could be clearer.
2. Consistency in terminology should be maintained, such as using "video analysis" or "video analytics" consistently.
3. The paper lacks detailed descriptions of state-of-the-art methods, making direct comparisons difficult.
4. Additional ablation experiments or visualizations of the model selection method could help readers understand the approach better, along with clearer explanations of some figures and experimental results.
5. The complexity of the AxiomVision framework, especially in managing dynamic model selection across numerous edge devices, could pose scalability challenges due to computational overhead and coordination requirements.
6. More introduction about the graph network and the rationale for its selection is needed. The mathematical proofs in Section 5 could also be moved to the appendix to avoid interfering with the main system design description.

In all, I thinks this paper can be accepted as a poster.